# Randomised controlled trial of automated VR therapy to improve positive self-beliefs and psychological well-being in young people diagnosed with psychosis: a study protocol for the Phoenix VR self-confidence therapy trial

Daniel Freeman [1,2] Jason Freeman,[1] Aitor Rovira,[1,2] Andre Lages Miguel,[1] Rupert Ward,[1,2] Matthew Bousfield,[1] Ludovic Riffiod,[1] Jose Leal [3] Thomas Kabir [1] Ly-Mee Yu,[4] Helen Beckwith,[1,2] Felicity Waite [1,2] Laina Rosebrock[1,2]

For numbered affiliations see end of article.

**Correspondence to**
Professor Daniel Freeman;
daniel.freeman@psy.ox.ac.uk

## ABSTRACT

**Introduction** The confidence of young people diagnosed with psychosis is often low. Positive self-beliefs may be few and negative self-beliefs many. A sense of defeat and failure is common. Young people often withdraw from many aspects of everyday life. Psychological well-being is lowered. Psychological techniques can improve self-confidence, but a shortage of therapists means that very few patients ever receive such help. Virtual reality (VR) offers a potential route out of this impasse. By including a virtual coach, treatment can be automated. As such, delivery of effective therapy is no longer reliant on the availability of therapists. With young people with lived experience, we have developed a staff-assisted automated VR therapy to improve positive self-beliefs (Phoenix). The treatment is based on established cognitive behavioural therapy and positive psychology techniques. A case series indicates that this approach may lead to large improvements in positive self-beliefs and psychological well-being. We now aim to conduct the first randomised controlled evaluation of Phoenix VR.

**Methods and analysis** 80 patients with psychosis, aged between 16 and 30 years old and with low levels of positive self-beliefs, will be recruited from National Health Service (NHS) secondary care services. They will be randomised (1:1) to the Phoenix VR self-confidence therapy added to treatment as usual or treatment as usual. Assessments will be conducted at 0, 6 (post-treatment) and 12 weeks by a researcher blind to allocation. The primary outcome is positive self-beliefs at 6 weeks rated with the Oxford Positive Self Scale. The secondary outcomes are psychiatric symptoms, activity levels and quality of life. All main analyses will be intention to treat.

**Ethics and dissemination** The trial has received ethical approval from the NHS Health Research Authority (22/LO/0273). A key output will be a high-quality VR treatment for patients to improve self-confidence and psychological well-being.

**Trial registration number** ISRCTN10250113.

## STRENGTHS AND LIMITATIONS OF THIS STUDY

⇒ This is a randomised controlled trial that will provide an estimate of the treatment effect of the addition of the Phoenix virtual reality self-confidence therapy to standard care.
⇒ The primary outcome is a new self-report measure of positive self-beliefs that is strongly associated with psychological well-being.
⇒ The secondary outcomes cover a wide range of psychiatric, functioning and quality-of-life domains.
⇒ The trial will not be able to determine the components of the intervention that lead to any clinical benefits.
⇒ The trial is only powered to detect at least moderate effect size improvements in positive self-beliefs.

## INTRODUCTION

Patients diagnosed with schizophrenia (and related non-affective psychosis) have often experienced difficult lives;[1] may be coping with distressing psychotic experiences, low mood and anxiety;[2] and can suffer stigma both from themselves and others.[3] Patients typically view themselves negatively, believe that they are inferior to others and can be overly self-critical.[4] Positive self-views that could counteract such beliefs are often not formed, consolidated or accessible.[5] The negative self-views, and limited levels of positive self-views, can exacerbate negative affect and psychotic experiences.[6] An end result is that often individuals withdraw from many aspects of their lives.[7] They may become inactive and isolated,[8] with further negative

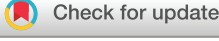

BMJ

consequences for mental and physical health. Overall, levels of psychological well-being are often very low in patients with psychosis.[9] We set out to develop a potentially scalable intervention that can build up positive self-beliefs and hence psychological well-being. Our aim is to arrest this downward spiral as soon as possible by focusing on patients at the early stages of psychosis.

Psychological techniques delivered in face-to-face therapy, and drawing on cognitive behavioural therapy and positive psychology,[10–12] can improve self-confidence. In a pilot randomised controlled trial of six sessions of face-to-face therapy with patients with current persecutory delusions in the context of a diagnosis of psychosis, we showed that it is possible to achieve large improvements in positive beliefs about the self, social comparison and psychological well-being.[13] However, very few patients receive such psychological therapy. There is a shortage of psychological therapists for patients with schizophrenia. Even fewer therapists have the time to take patients into real-world situations where the most important work is conducted: building up self-confidence through direct experience. Our potential solution is to use immersive technology. Virtual reality (VR) therapy involves wearing a headset and interacting with computer-generated simulations in carefully selected scenarios. The significant advantages of VR are that: it allows users repeatedly to experience therapeutically beneficial simulations; learning has been demonstrated to transfer from the virtual environment to the real world;[14–16] the therapy can be made compelling and entertaining; and treatment can be automated within VR, removing reliance on the availability of therapists.[17]

In a user-centred design process with young people with lived experience of psychosis we designed a VR therapy to improve positive self-beliefs (Phoenix VR) (Rosebrock *et al*, submitted). The treatment built on our face-to-face therapy. The VR therapy had high usability ratings. The treatment is programmed to run on a stand-alone VR consumer headset that can be left with patients. A staff member then has up to six sessions with the patient to introduce the VR therapy and help the person apply the learning to the real world. Patients can have these sessions at home or in the clinic. An initial proof of concept case series test of Phoenix VR therapy was conducted.[18] Twelve young patients diagnosed with psychosis received Phoenix. There were large improvements in positive beliefs about the self (d=3.0) and psychological well-being (d=1.5) after the intervention. However, this evaluation did not include a control group. As such, improvements could have been a consequence of natural recovery. A randomised controlled evaluation of Phoenix VR self-confidence therapy is now required.

## Aims and hypotheses

Our primary research question is: Does Phoenix VR self-confidence therapy added to treatment as usual, compared with treatment as usual alone, lead to a post-treatment improvement in levels of positive self-beliefs for patients at the early stages of psychosis attending NHS mental health services?

Our primary hypothesis is that:

Compared with treatment as usual, Phoenix VR self-confidence therapy added to treatment as usual will improve levels of positive self-beliefs (end of treatment).

Our secondary hypotheses are as follows:

1. Compared with treatment as usual, Phoenix VR self-confidence therapy added to treatment as usual will improve social comparison and psychological well-being (end of treatment).
2. Compared with treatment as usual, Phoenix VR self-confidence therapy added to treatment as usual will reduce depression, negative self-beliefs, hopelessness, anhedonia, anxiety, and paranoia and increase perceptions of recovery, meaningful activity and quality of life (end of treatment).
3. Treatment effects will be maintained at follow-up (3 months).

We will also collect data on satisfaction, side effects and health economic variables.

## METHODS AND ANALYSIS
### Trial design and flow chart

The design is a parallel group, superiority, single-blind, randomised controlled trial to test the effects of Phoenix VR therapy when added to usual care. The trial will be run from a single centre, with up to four NHS mental health trusts participating (Oxford Health NHS Foundation Trust; Berkshire Healthcare NHS Foundation Trust; Northamptonshire Healthcare NHS Foundation Trust; Central and North West London NHS Foundation Trust (Milton Keynes)). Treatment as usual will be measured but remain unchanged in both groups. Assessments will be conducted at 0, 6 and 12 weeks. Assessments will be conducted in person or online by research assistants blind to group allocation. A summary of the trial design can be seen in figure 1. The trial is prospectively registered with the ISRCTN registry: ISRCTN10250113. The University of Oxford is the trial sponsor. There is a data monitoring and ethics committee (DMEC).

### Randomisation, blinding and code-breaking

Participants will be randomised after completing the baseline assessment and allocated to one of the trial arms using a 1:1 allocation ratio. Randomisation will be carried out by a validated online system provided by Sealed Envelope (www.sealedenvelope.com) and using a permuted blocks algorithm, with randomly varying block size.

The research assessors will be blinded to group allocation, but the patients and staff member present will not (they cannot be blinded to whether psychological intervention is delivered or received). If an allocation is revealed between assessment sessions, this will be logged by the trial coordinator and reblinding will occur using another assessor.

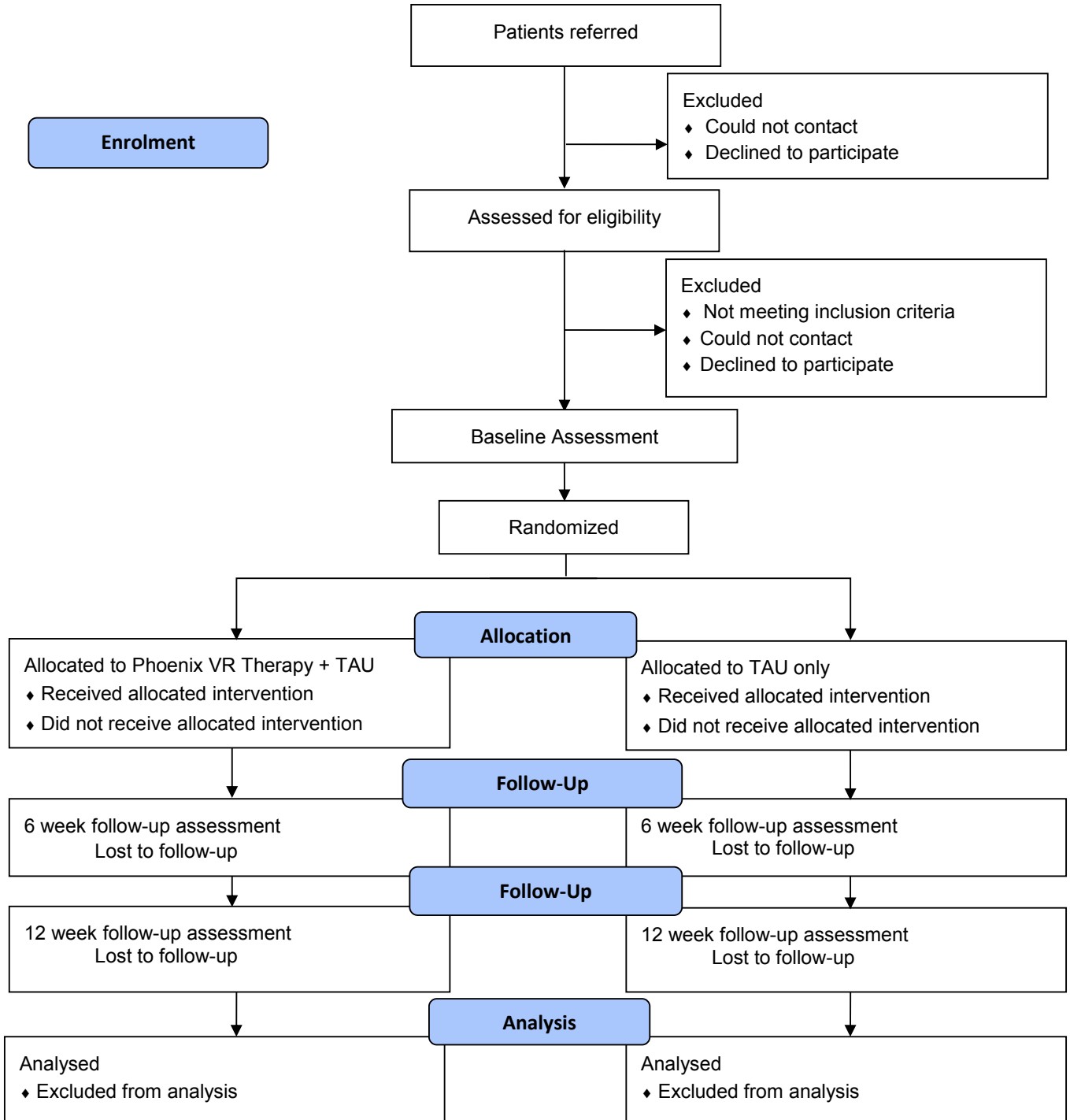

**Figure 1** Trial flow diagram. VR, virtual reality; TAU, treatment as usual.

## Participants

Participants will be NHS patients diagnosed with non-affective psychosis, with low positive self-beliefs, and aged 16–30 years old. Participants will be recruited principally by seeking referrals to the trial from the relevant clinical teams in the participating mental health trusts (early intervention services, adult community mental health teams and inpatient units). Patients interested in taking part will then be approached by the research team with the approval of the clinical team, given information about the trial and screening conducted. All suitable patients will be given at least 24 hours to consider taking part in the trial. Our Lived Experience Advisory Panel (LEAP) has emphasised the importance of patients being able to self-initiate referral. This minimises the chance that particular patients miss out on trials because they are overlooked by clinical teams or their clinician was absent from a referral meeting. However, in all instances, we will seek to confirm

that a member of the clinical team gives approval for a patient to enter the trial and to complete the necessary screening of eligibility and risk status. Written informed consent will be obtained from all participants.

The inclusion criteria are: Participant is willing and able to give informed consent for participation in the trial; aged between 16 years and 30 years old; attending NHS mental health services for treatment of psychosis; primary clinical diagnosis of non-affective psychosis (schizophrenia, schizoaffective disorder, delusional disorder or psychosis NOS); low levels of positive self-beliefs indicated by a score of 50 or lower on the Oxford Positive Self Scale;[19] stable medication for at least 1 month and no planned significant changes at the outset of participation.

The exclusion criteria are: a primary diagnosis of another mental health condition (eg, substance use disorder) that would be the first clinical priority to treat; current engagement in any other intensive individual psychological therapy; in forensic settings or psychiatric intensive care unit (PICU); command of spoken English inadequate for engaging in the VR therapy; photosensitive epilepsy or significant visual, auditory or balance impairment that would make use of VR inappropriate; significant learning difficulties that would prevent the completion of assessments. A participant may also be excluded if there is another factor (eg, current active suicidal plans) which, in the judgement of the investigator, would preclude the participant from providing informed consent or from safely engaging with the trial procedures. Reason for exclusions will be recorded.

## Assessments

The primary outcome is the total score on the Oxford Positive Self Scale.[19] This is a 24-item self-report scale assessing four types of positive self-belief: mastery, strength, enjoyment and character. It was developed with a representative sample of 2500 members of the UK population, with further validation in another three thousand individuals.

It has a high correlation with psychological well-being (r=0.79). The psychometrics of the scale are excellent, including internal consistency (omega coefficient=0.92).

Secondary outcomes are assessments of social comparison, specifically the Social Comparison Scale,[20] and psychological well-being, using the Warwick-Edinburgh Well-being Scale.[21] Negative and positive self-beliefs will also be assessed with the Brief Core Schema Scale-Negative and Positive Self subscales[5] and the Everyday Confidence Scale.[22] Depression will be assessed with the Patient Health Questionnaire-9,[23] hopelessness with the Beck Hopelessness Scale,[24] anhedonia with the Temporal Experience of Pleasure Scale-anticipatory pleasure,[25] anxiety with the GAD-7 (Generalised Anxiety Disorder Assessment)[26] and paranoia with the Revised Green *et al* Paranoid Thoughts Scale.[27] We will also assess perceptions of recovery with the Questionnaire of Process of Recovery,[28] meaningful activity with a time budget,[29] and quality of life with the EQ-5D-5L (EuroQol)[30] and the ReQol.[31] Service use data will be collected with the Modified Client Service Receipt Inventory.[32] For patients in the Phoenix VR therapy arm, the staff member will collect end-of-treatment assessments of satisfaction, using the Modified Client Satisfaction Questionnaire,[33 34] and side effects, via the Oxford-VR Side Effects Scale.[35] A summary of the measures is provided in table 1.

### The VR psychological treatment

The treatment being tested is Phoenix VR self-confidence therapy, which is a VR application recommended for adults (16+) attending psychosis services who have low levels of positive self-beliefs. This software is intended to increase positive self-beliefs. The treatment was designed by our team at the University of Oxford, with young people with lived experience taking part in the design process. The treatment was programmed by the University of Oxford. Phoenix is a UKCA marked, class I medical device (standalone software as a medical device).

**Table 1** Summary of objectives and assessment measures

| | Objectives | Outcome measures |
|---|---|---|
| Primary | Test whether the VR treatment leads to improvement in positive self-beliefs. | Oxford Positive Self Scale |
| Secondary | Test whether the VR therapy improves social comparison and psychological well-being. | Social Comparison Scale, Warwick-Edinburgh Well-being Scale |
| | Test whether the VR therapy leads to reductions in depression, negative self-beliefs, everyday self-confidence, hopelessness, anhedonia, anxiety, and paranoia and increases in perceptions of recovery, meaningful activity and quality of life. | Patient Health Questionnaire-9, Brief Core Schema Scale, Everyday Confidence Scale, Beck Hopelessness Scale, Temporal Experience of Pleasure Scale, Generalised Anxiety Disorder Assessment (GAD-7), Revised Green *et al* Paranoid Thoughts Scale, Questionnaire of Process of Recovery, time budget, EQ-5D-5L (EuroQol), ReQol. |
| | Assess patient satisfaction with the VR therapy. | Modified version of the Client Satisfaction Questionnaire. |
| | Assess side effects of the VR therapy | Oxford-VR Side Effects Scale |

VR, virtual reality.

The Phoenix software application is composed of a set of virtual environments, including different scenes created using three-dimensional (3D) models, ambient audio and 3D virtual characters, with animations and speech. The environments are driven by source code which handles the logic of the programme, the behaviour of the computer characters, as well as the user interaction and data storage. The software is built using Unity (Unity Technologies). Unity acts as a render engine, displaying the virtual environments to the user through the headset. The application runs through the Unity software application on a Meta Quest 2 VR Headset.

The primary treatment goal is to help young people who have been given a diagnosis of schizophrenia (or related condition) to increase their psychological well-being by building up positive self-beliefs. Patients can keep the headset for 6 weeks in order to use Phoenix on their own. The treatment is supported by a staff member over six sessions. In the first meeting, the staff member introduces Phoenix and helps the person try it out. The staff member also helps the person set real-world goals and between-session tasks to improve confidence and reviews progress. VR does not need to be used in the subsequent in-person meetings since the patient is able to use it at home but most sessions to date have included approximately twenty minutes within VR. Phoenix is used to spark positive self-beliefs that are then consolidated via real-world activities.

The six key mechanisms of action within Phoenix are to: build vivid memories of successful and rewarding interactions; help identify strengths; learn to tolerate fearful emotions and still succeed; learn to appraise situations more positively and with more self-kindness; develop the ability to savour positive experiences and connect with other people; and increase engagement in meaningful activities. Phoenix has three main therapeutic areas within VR for the user: creating a sense of achievement and mastery to develop the belief 'I can make a difference'; succeeding in challenging situations to develop the belief 'I can do this'; and engaging in pleasurable activities to develop the belief 'I can enjoy things'. The achievement and mastery scenario is set in a community garden, with ten tasks relating to the care of plants, animals and a farmhouse. Each task is broken down into several steps for the participant to complete. For example, caring for plants includes preparing the soil, planting seeds, watering and picking the crops. The challenging situation is a TV studio, with the user having to speak to camera in front of an audience. Patients read out a weather forecast presented on an autocue. There are 10 levels of difficulty, varying in the size of the audience, the difficulty of the speaking task, and the environment itself (eg, gradual introduction of cameras and spotlights). The pleasurable experiences are set near a lake by a forest and tasks include relaxation exercises (eg, savouring, progressive muscle relaxation) and games (eg, a throwing game, playing on a drum set). A virtual coach (called Farah) guides the participant in the best way to think, feel and respond. Farah provides instruction, encouragement and positive reinforcement. Farah first meets users in her glass-fronted office that overlooks mountains and provides instructions throughout the programme. Users choose which VR scenarios they wish to complete and can repeat activities as often as they would like.

## Control condition

Participants who are allocated to the control arm will continue to receive their usual care. No additional interventions will be offered by the research team. Treatment as usual for the participants within this trial will typically consist of prescription of psychiatric medications and meetings with a mental health practitioner. Treatment as usual will vary across individuals. We will collect detailed data on treatment as usual.

## Adverse events

A trial standard operational procedure has been written for adverse events. We will record the occurrence of any serious adverse events (SAE) reported to us and also check each patient's medical notes at the end of their participation in the trial. An adverse event is defined by the ISO14155:2011 guidelines for medical device trials as serious if it: (A) results in death, (B) is a life-threatening illness or injury, (C) requires hospitalisation or prolongation of existing hospitalisation, (D) results in persistent or significant disability or incapacity, (E) medical or surgical intervention is required to prevent any of the above, (F) leads to foetal distress, foetal death or consists of a congenital anomaly or birth defect or (G) is otherwise considered medically significant by the investigator.

Life-threatening in the definition of an SAE refers to an event in which the subject was at risk of death at the time of the event; it does not refer to an event that hypothetically might have caused death if it were more severe. A planned hospitalisation for a pre-existing condition, without a serious deterioration in health, is not considered to be an SAE.

The sorts of SAEs that can typically happen to this participant group include: deaths, suicide attempts, serious violent incidents and admissions to hospital. SAEs are recorded using an SAE report form. The relationship between the investigational medical device or other research procedure and the occurrence of each SAE will be assessed and categorised. The investigators will use clinical judgement to determine the relationship (including whether the patient reports a link to the study). Alternative causes, such as natural history of the participant's underlying condition, concomitant therapy, other risk factors, etc, will be considered. The study team will make an initial assessment of whether the SAE is potentially related to the device or trial procedures and report to the regulatory authorities within the appropriate timescales. The decision about relatedness will be reviewed by the DMEC chair in the first instance.

We will also record adverse events that are not serious. This would include any adverse device effects

from the VR treatment, including those resulting from insufficient or inadequate instructions for use, deployment, installation or operation, or any malfunction of the software. It also includes any event resulting from user error or intentional misuse.

## Analysis

A full-statistical analysis plan will be approved before any analysis. We will report data in line with the CONSORT (Consolidated Standards of Reporting Trials) 2010 statement[36] showing attrition rates and loss to follow-up. The primary analyses will be carried out using the intention-to-treat principle. That is, after randomisation, participants will be analysed according to their allocated intervention arm irrespective of what intervention they actually receive, and with data available from all participants included in the analysis including those who do not complete therapy. The outcome analyses will be conducted by statisticians in the University of Oxford Primary Care Clinical Trials Unit.

Baseline variables will be presented by randomised group using frequencies (with percentages) for binary and categorical variables and means (and SD) or medians (with lower and upper quartiles) for continuous variables, along with minimum and maximum values and counts of missing values. There will be no tests of statistical significance nor confidence intervals for differences between groups on any baseline variables. There will be no planned interim analysis for efficacy or futility on the primary outcome.

We will test the primary hypothesis for between-group difference in the primary outcome using a linear mixed effects model which models the response at 6 weeks and 12 weeks, with baseline outcome measure and treatment assignment as fixed effects, with a patient specific random intercept. An interaction between time and randomised group will be fitted as a fixed effect to allow estimation of treatment effect at all time points. The linear mixed effects model will account for missing data assuming data are missing-at-random. Standard residual diagnostics will be assessed for the appropriateness of the model. A $p < 0.05$ will be used as the level of statistical significance. Similar mixed effect models will be used to analyse secondary outcomes. Treatment differences estimated from linear mixed effects models will additionally be reported as standardised mean differences (mean group difference divided by whole group SD at baseline).

The sample size of 80 patients takes into consideration a maximum attrition rate of 10% and provides 80% power to detect a moderate difference (d=0.67) in positive self-beliefs, from randomisation to 6 weeks at a 5% level of significance using an independent groups t-test (two sided). Further power is likely to be gained by use of mixed-effects models.

## Ethics and dissemination

The trial has received Health Research Authority (HRA) approval (IRAS 312539). The trial received ethical approval from the NHS London—Harrow Research Ethics Committee (22/LO/0273). The results of the trial will be published in a peer-reviewed journal and made open access. Deidentified participant data will be available in anonymised form on reasonable request, subject to review and contract with the University of Oxford, following the publication of results.

## Patient and public involvement

Patient and public involvement has been supported by the McPin Foundation, a charity that aims to put the lived experience perspective at the heart of mental health research. A patient advisory group (PAG), comprising young people who have been diagnosed with psychosis, has helped design the VR therapy, try prototypes and contributed to the development of the primary outcome measure.

**Author affiliations**
[1]Department of Experimental Psychology, University of Oxford, Oxford, UK
[2]Oxford Health NHS Foundation Trust, Oxford, UK
[3]Health Economics Research Centre, Nuffield Department of Population Health, Oxford, UK
[4]Oxford Primary Care Clinical Trials Unit, Nuffield Department of Primary care Health Sciences, University of Oxford, Oxford, UK

**Contributors** DF is the chief investigator, conceived the project, secured the funding, had overall responsibility for the treatment design and the trial design and drafted the trial protocol. LRo, JF, FW, AR, ALM, RW, MB and LRi contributed to the treatment design. AR, ALM, RW, MB and LRi conducted the programming. LRo is co-ordinating the trial. TK is responsible for lived experience involvement. L-MY is responsible for the statistical analysis. JL will be responsible for health economic data. All authors commented on the trial protocol.

**Funding** International Foundation (https://www.if-internationalfoundation.org/en/home-page) award to DF. The project is also supported by the NIHR Oxford Health Biomedical Research Centre. DF is an NIHR Senior Investigator and is supported by the NIHR Oxford Health Biomedical Research Centre. FW is supported by a Wellcome Trust Doctoral Fellowship.

**Disclaimer** The views expressed are those of the authors and not necessarily those of the NHS, the NIHR or the Department of Health.

**Competing interests** DF and JF are founders of Oxford VR, a University of Oxford spin-out company. Oxford VR has not been involved in the Phoenix project.

**Patient and public involvement** Patients and/or the public were involved in the design, or conduct, or reporting, or dissemination plans of this research. Refer to the Methods section for further details.

**Patient consent for publication** Not applicable.

**Provenance and peer review** Not commissioned; externally peer reviewed.

**ORCID iDs**
Daniel Freeman http://orcid.org/0000-0002-2541-2197
Jose Leal http://orcid.org/0000-0001-7870-6730
Thomas Kabir http://orcid.org/0000-0001-8908-0964

Felicity Waite http://orcid.org/0000-0002-2749-1386

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
