## [Reviewer comments · BMJ Open]

ARTICLE DETAILS

TITLE (PROVISIONAL)	A Randomised Controlled Trial of Automated VR Therapy to Improve Positive Self-Beliefs and Psychological Wellbeing in Young People Diagnosed with Psychosis: Study Protocol for the Phoenix VR Self-Confidence Therapy Trial
AUTHORS	Freeman, Daniel; Freeman, Jason; Rovira, Aitor; Miguel, Andre Lages; Ward, Rupert; Bousfield, Matthew; Riffiod, Ludovic; Leal, Jose; Kabir, Thomas; Yu, Ly-Mee; Beckwith, Helen; Waite, Felicity; Rosebrock, Laina

VERSION 1 – REVIEW

REVIEWER	Ruini, Chiari University of Bologna
REVIEW RETURNED	05-Sep-2023

GENERAL COMMENTS	This paper introduces a study protocol that aims to evaluate the efficacy of a Virtual Reality Self-Confidence Therapy Trial in enhancing positive self-beliefs and psychological well-being among adolescents diagnosed with psychosis. The articulation of the aim and hypothesis is both clear and well-justified. The exposition of the reasoning underlying each choice is comprehensive, and the outlined statistical analysis and experimental design exhibit robust methodological underpinnings. In summary, the research protocol is eloquently composed, and the proposed methodology aligns with the study's objectives. My feedback primarily pertains to two minor facets: a recommendation to bolster assertions with relevant citations and an inquiry seeking additional info. Given the expected minor nature of these suggested refinements, I am inclined to suggest categorizing this article as requiring "Minor Revisions." Minor feedback: 1. Within the introduction, the authors make the assertion that learning can be translated effectively from the virtual environment to the physical realm. To substantiate this claim, I propose the inclusion of supporting citations.2. In connection with the Oxford Positive Self Scale, it would be prudent to include the Cronbach's alpha coefficient to bolster the validation of the selected scale.3. While the activities proposed within the VR environment are comprehensively outlined, I propose a more comprehensive explication of the rationale underlying each task to enhance the reader's understanding.
---

REVIEWER	Pillny, Matthias Universität Hamburg, Psychology
-----------------	---

REVIEW RETURNED	20-Sep-2023
-------------

GENERAL COMMENTS	Thank you for inviting me this fantastic manuscript. The study protocol is well written with a clear and concise description of the study rationale. I have only some minor comments on the study methodology that I would like to share for the author's consideration and which might help to improve the methodological rigour of the study:  1. Selection of participants: I welcome the possibility for patients to put themselves forward for the study. However, I wonder if this could lead to the exclusion of participants who would otherwise be eligible? If so, I can see the possibility that this may bias the selection of participants (e.g. only those for whom the approach is most appropriate will be selected). Also, I can imagine that some interested and eligible participants may feel patronised if a doctor refuses them access to the study they want to participate in. Is there a way to include all eligible participants, regardless of the clinician's subjective assessment? 2. Classification of adverse reactions: I suggest to ask participants first whether they feel an association between Phoenix and an adverse event as a primary indicator of relatedness. Clinical judgement could be used for a secondary indicator. It is fine to prefer the clinical judgement in case of non convergence, but I would always assess the participant's opinion. 3. Test for differences at baseline: Although this is a randomised trial, I believe that formal testing of differences (at least in some) baseline characteristics would be informative in determining whether randomisation was successful and, in the case of significant differences, could inform additional analyses could be conducted testing these differences as covariates. 4. Do the authors plan to impute missing data? 5. As the authors state that they will include the group x time interaction as fixed effect in their model, I am a bit surprised why they chose to include the baseline values as a separate fixed effect? Intuitively, I would also model the baseline values in the group x time interaction term to reduce the number of predictors in the model. Could you explain your rationale for this? 6. Can the authors please describe how the sample size was determined in more detail? 7. Could the authors add one or two sentences explaining their rationale for using the SD at baseline to calculate the SMDs?
--

VERSION 1 – AUTHOR RESPONSE

Reviewer: 1

Dr. Chiari Ruini, University of Bologna

This paper introduces a study protocol that aims to evaluate the efficacy of a Virtual Reality Self-Confidence Therapy Trial in enhancing positive self-beliefs and psychological well-being among adolescents diagnosed with psychosis. The articulation of the aim and hypothesis is both clear and well-justified. The exposition of the reasoning underlying each choice is comprehensive, and the outlined statistical analysis and experimental design exhibit robust methodological underpinnings.

In summary, the research protocol is eloquently composed, and the proposed methodology aligns with the study's objectives. My feedback primarily pertains to two minor facets: a recommendation to bolster assertions with relevant citations and an inquiry seeking additional info. Given the expected minor nature of these suggested refinements, I am inclined to suggest categorizing this article as requiring "Minor Revisions."

Response: We thank the reviewer for the kind comments, which are appreciated.

Minor feedback:

1. Within the introduction, the authors make the assertion that learning can be translated effectively from the virtual environment to the physical realm. To substantiate this claim, I propose the inclusion of supporting citations.

Response: This has been added.

2. In connection with the Oxford Positive Self Scale, it would be prudent to include the Cronbach's alpha coefficient to bolster the validation of the selected scale.

Response: Thank you. We have added the omega coefficient for internal consistency.

3. While the activities proposed within the VR environment are comprehensively outlined, I propose a more comprehensive explication of the rationale underlying each task to enhance the reader's understanding.

Response: Thank you. We have added additional detail on the tasks.

Reviewer: 2
Dr. Matthias Pillny, Universität Hamburg

Thank you for inviting me this fantastic manuscript. The study protocol is well written with a clear and concise description of the study rationale. I have only some minor comments on the study methodology that I would like to share for the author's consideration and which might help to improve the methodological rigour of the study:

Responding: We really appreciate the positive response. Thank you.

1. Selection of participants: I welcome the possibility for patients to put themselves forward for the study. However, I wonder if this could lead to the exclusion of participants who would otherwise be eligible? If so, I can see the possibility that this may bias the selection of participants (e.g. only those for whom the approach is most appropriate will be selected). Also, I can imagine that some interested and eligible participants may feel patronised if a doctor refuses them access to the study they want to participate in. Is there a way to include all eligible participants, regardless of the clinician's subjective assessment?

Response: We are keen for as many different routes for patients in services to enter the study. The reality is that far fewer patients self-refer (since many will not see any advertising). We have never had an instance of a patient self-refer, meet the entry criteria, and have the responsible clinical team refuse agreement. Clinically we should follow the guidance of the responsible clinical team, since they would clearly have a good reason if such a circumstance arose. We will record if any patients self-refer, are suitable, and then the team refuse agreement.

2. Classification of adverse reactions: I suggest to ask participants first whether they feel an association between Phoenix and an adverse event as a primary indicator of relatedness. Clinical judgement could be used for a secondary indicator. It is fine to prefer the clinical judgement in case of non convergence, but I would always assess the participant's opinion.

Response: Thank you. Whether a patient considers there a link is always considered in the judgements (and this has now been added to the text). Of course in some instances it is not possible to obtain the patient's view.

3. Test for differences at baseline: Although this is a randomised trial, I believe that formal testing of differences (at least in some) baseline characteristics would be informative in determining whether randomisation was successful and, in the case of significant differences, could inform additional analyses could be conducted testing these differences as covariates.

Response: There should not be formal testing. Journals and best practice guidelines are rightly trying to stop what is sometimes called the "Table 1 Fallacy". The reasons are well set out here: Senn, S. (1994). Testing for baseline balance in clinical trials. *Statistics in medicine*, 13(17), 1715-1726.

4. Do the authors plan to impute missing data?

Response: We do not plan to do this. We are using mixed effect models, which are more efficient under the missing at random mechanism. Our trials have very low drop-out in any case. However if there was significant missing drop out we would conduct sensitivity tests.

5. As the authors state that they will include the group x time interaction as fixed effect in their model, I am a bit surprised why they chose to include the baseline values as a separate fixed effect? Intuitively, I would also model the baseline values in the group x time interaction term to reduce the number of predictors in the model. Could you explain your rationale for this?

Response: Baseline is a covariate before randomisation rather than an outcome after randomisation, so should be fitted as a fixed effect. We are not building a prediction model.

6. Can the authors please describe how the sample size was determined in more detail?

Response: Thank you. Further detail has been added.

7. Could the authors add one or two sentences explaining their rationale for using the SD at baseline to calculate the SMDs?

Response: There are pros and cons for using the SD at baseline or at follow-up. The obvious advantage of baseline is that it is from the whole population suitable for the trial before any further variability is added via treatment or there is loss to drop-out (i.e. a representativeness point) (which is why we favour it). However, whichever researchers choose, the most important point is to specify in advance, which is exactly what we do.